# Sustainability of a Multi-Component Education Program (ABC of Healthy Eating) after Three Months and Nine Months: The Socioeconomic Context in Improving Nutrition Knowledge in Polish Teenagers

**DOI:** 10.3390/nu13051661

**Published:** 2021-05-14

**Authors:** Lidia Wadolowska, Malgorzata Kostecka, Joanna Kowalkowska, Marta Jeruszka-Bielak, Marzena Tomaszewska, Anna Danielewicz, Jadwiga Hamulka

**Affiliations:** 1Department of Human Nutrition, Faculty of Food Science, University of Warmia and Mazury in Olsztyn, Pl. Cieszynski 1, 10-718 Olsztyn, Poland; lidia.wadolowska@uwm.edu.pl (L.W.); joanna.kowalkowska@uwm.edu.pl (J.K.); anna.danielewicz@uwm.edu.pl (A.D.); 2Department of Chemistry, Faculty of Food Science and Biotechnology, University of Life Sciences, 15 Akademicka Street, 20-950 Lublin, Poland; 3Department of Human Nutrition, Institute of Human Nutrition Sciences, Warsaw University of Life Science (SGGW-WULS), 159C Nowoursynowska Street, 02-787 Warsaw, Poland; marta_jeruszka_bielak@sggw.edu.pl (M.J.-B.); jadwiga_hamulka@sggw.edu.pl (J.H.); 4Department of Food Gastronomy and Food Hygiene, Institute of Human Nutrition Sciences, Warsaw University of Life Science (SGGW-WULS), 159C Nowoursynowska Street, 02-787 Warsaw, Poland; marzena_tomaszewska@sggw.edu.pl

**Keywords:** adolescent, education, intervention, nutrition knowledge, social factors

## Abstract

The study aimed to evaluate the sustainability of a multi-component education (ABC-HEat) program related to healthy nutrition and lifestyle after three months and nine months and to assess the socioeconomic context in improving teenage nutrition knowledge. The study was designed as a clustered, controlled, education-based intervention. A sample was chosen and allocated into either an educated group (under intervention) or a control group (outside of intervention). The study covered 464 11–12-year-old students (educated/control 319/145). In the educated group, data were collected three times: before education, after three months and after nine months to measure the short- and the long-term effects of education, respectively. In the control group, data were collected in parallel. Changes in nutrition knowledge score (NKS, points) by sex, residence, family affluence scale (FAS) were the main outcome measures. The increase in the NKS was significantly higher in the educated group than in the control group—three months after education on average by 1.4 to 2.7 points (all *p* < 0.001) in the total sample and all subgroups, and nine months after education in rural residents by 2.2 points (*p* < 0.001) and in the total sample by 0.4 (*p* < 0.05). In the educated group, the chance of no increase in the NKS was higher in urban than rural residents after three months and nine months (adjusted odds ratios [OR] and 95% Confidence Intervals [95% CI]: 3.63, 1.80–7.31 and 2.99, 1.60–5.59, respectively, both *p* < 0.001) using the increase in the NKS by ≥4 points as a reference. The multi-component education program improved the nutrition knowledge of teenagers in the short term regardless of socioeconomic variables, but in the long term this effect was visible only in rural residents. It suggests that a special path of nutrition education addressed to urban teens may be required.

## 1. Introduction

Nutrition knowledge forms the basis for shaping attitudes towards foods, nutrition and health and human dietary behaviours [1,2,3]. Nutrition knowledge, through its direct influence on dietary behaviours, can be considered as an important factor that indirectly influences human health. Dietary behaviours are formed in early childhood [4,5] and change throughout the lifespan, but key habits and food preferences are relatively stable over time, e.g., craving for sweets or a tendency to over-consume fats [6,7,8]. It follows that changing dietary behaviours and developing nutrition knowledge should take place in childhood and adolescence before undesirable dietary habits become established.

The effectiveness of nutrition education depends on many factors, including the content of the program, its goals and duration and the method used that should be adapted to the developmental age of respondents, their perceptive abilities and needs [9,10,11]. Comparative studies have shown that multi-component intervention is more effective than single-component intervention [12,13,14,15], especially if it includes a practical component (e.g., workshops) combined with theoretical messages. In the Canadian program “Action Schools! BC—Healthy Eating”, which used a multi-component approach with educational, environmental and family elements, a significant increase in the consumption of fresh vegetables and fruits was found [16]. This finding was confirmed in later studies [15,17,18,19]. It seems that the success of the program, apart from its complexity, may depend on the complementarity of the program components, as well as the proper identification of health benefits by young people resulting from the adoption of pro-healthy dietary habits [20,21].

The influence of socioeconomic factors on dietary and lifestyle behaviours is well known [22,23,24]. Many studies show that young people from families with lower socioeconomic status consume less pro-healthy foods such as vegetables and fish [1,25,26,27], while more foods with lower nutrient contents but higher energy value, e.g., fast foods [28,29,30]. It has been suggested that the lower socioeconomic status of the family is related to the lower interest of teenagers in a pro-healthy diet [31,32]. Therefore, it can be speculated that family socioeconomic status, determined with parents’ education level, family affluence and urban vs. rural residence, is an important factor influencing the nutrition knowledge level in young people [33]. It has been shown that a poorer social environment in which young people grow up can have a devastating effect on nutrition education [34]. Therefore, socioeconomic status may determine the effectiveness of educational programs targeted at adolescents.

The impact of socioeconomic status on the nutrition knowledge level of young people and the effectiveness of nutrition- and health-related education programs has not been known so far. Filling this knowledge gap will identify inequalities in the long-term maintenance of nutrition knowledge and identify groups of children and adolescents that require a special approach when education programs are designed. The aim of the current study was to evaluate the sustainability of a multi-component education program related to healthy nutrition and lifestyle after three months and nine months and to assess the socioeconomic context in improving teenage nutrition knowledge.

## 2. Materials and Methods

### 2.1. Study Design and Participants

The data for this study were collected in Poland in 2015–2016 as a part of the “ABC of Healthy Nutrition” project, the 1st edition of the national multicentre “ABC of Healthy Eating” (ABC-HEat) project (2015–2018) [35]. Data were collected by experienced researchers well trained in collecting dietary data.

The study was designed as a clustered, controlled, education-based intervention. A sample was chosen and allocated (not randomly) into either an educated group (under educational intervention) or a control group (outside of educational intervention).

In the educated group, data were collected three times: (i) before education (at baseline), (ii) after three months (±2 weeks; 3 month follow-up) to measure the short-term effect of education, (iii) after nine months (±2 weeks; 9 month follow-up) to measure the long-term effect of education. In the control group, data were collected in parallel.

The study included students of selected elementary schools from urban, suburban and rural areas. The schools (not randomly selected) were located in eight locations covering the entire territory of Poland in line with the location of seven academic centres involved in the study (Appendix A).

A school class was the smallest unit in the sample selection. It was decided to start recruitment based on school classes, because students were subject to the same school education and would be at a similar stage of development. Schools were first invited to take part in the study and each student of 4th- and 5th-grade classes of this school (classes that met criteria and teachers who gave their approval) were then invited. More details on the study design and protocol were described previously by Hamulka et al. [35].

School inclusion criteria were:a location at a convenient distance from the academic centres (up to 50 km);the consent of the school principal to school participation;no previous participation of the school in other nutrition-health education programs.

Participant inclusion criteria were:written consent of parents or legal guardians to participate;4th- and 5th-grade classes of elementary school; the expected age of the students was 11–12 years at baseline;no disability self-declared by a parent, legal guardian or teacher.

In total, 48 classes from 14 schools were selected across Poland (Figure 1). Initially, 668 students were recruited. A total of 204 participants were excluded from analyses (17 participants because of age and 187 participants due to not attending all stages of the study). The study included 464 teenagers aged 11–12 years, 216 boys (46.6%) and 248 girls (53.4%) (Table 1).

The sample size included in the study (educated/control group 319/145) exceeded the minimum sample size (200/104), which was determined in regard to nutrition knowledge as a key measure of the study. The calculation was based on the expected increase in nutrition knowledge score (see Section 2.4) after a 9 month follow-up and the expected difference between the educated and control groups. With a 5% significance level and 80% power, the sample size required approximately 304 respondents (assuming a 2/1 ratio for the educated/control group, i.e., 200/104 respondents and an equal number of respondents in each research centre) at the end of the study, to detect a 50% difference between the educated group (an increase of 30%) and the control group (an increase of 15%) in the increase of nutrition knowledge score, including a 50% dropout rate (at the end of the study) and a recoding error or missing data rate of 10%.

### 2.2. Intervention

The participants of the educated group were taken over a multi-component education program lasting three weeks and covering five diet-related and lifestyle-related topics, which included various forms of education, from fun to “scientific” cognition. Each topic lasted approximately 180 min (4 h of school lessons) and was run by a minimum of 3–4 researchers. The program was developed and implemented by academic researchers. Most of the educational activities were carried out in academic centres and the rest in schools. School teachers were not involved in the education program, they were only present during educational activities. Apart from the study, students from both educated and control groups took part in regular school activities containing some content related to nutrition and a healthy lifestyle. The topics and details of the education program are presented in Appendix A and were described previously [35,36].

### 2.3. Data Collection

Data related to nutrition knowledge and socioeconomic variables (self-reported) were collected with a short form of a food frequency questionnaire (acronym: SF-FFQ4PolishChildren) dedicated to school-aged children (Figure 2). The questionnaire was developed for the “ABC of Healthy Eating” project and its internal compatibility was tested [37].

### 2.4. Nutrition Knowledge

The nutrition knowledge level was determined based on eighteen questions and is presented in the Appendix A [35,36]. The questions were developed based on a questionnaire described by Whati et al. [38] and adapted to Polish conditions and education [37]. Correct answers were scored with 1 point and wrong or “I don’t know” answers or missing data were scored with 0 points. Points were summed up for each respondent to calculate the nutrition knowledge score (NKS, in points; ranged from 0 to 18). While developing the NKS, the authors were inspired by similar scores previously described by other researchers and authors [38,39,40,41]. In the present study, the value of Cronbach’s alpha for an NKS of 0.685 can be interpreted as sufficient internal consistency of the score (acceptable internal consistency of a score/scale is ≥0.7 while results ≥0.6 have been also reported as satisfactory or sufficient) [42,43]. A correlation coefficient of 0.68 was found for the NKS when the test–retest reproducibility of the questionnaire (SF-FFQ4PolishChildren) was assessed in 11–15-year-old adolescents [37].

### 2.5. Socioeconomic Data

Respondents were distinguished as rural or urban residents, based on the data collected with the questionnaire. The socioeconomic status was determined with the Family Affluence Scale (FAS) based on household characteristics (the details were described previously) [35,37]. For international use, the scale was developed with the Health Behaviour in School-aged Children (HBSC) cross-national study. For national use, the scale was adopted by the Polish team of the HBSC study [44]. The scale was composed of four questions related to having a family owning a car/van/truck, traveling away on holiday with the family, having a bedroom, having a family which owns computers/laptops/tablets, with answers to choose from. For each answer, points were assigned (Appendix A). To calculate FAS, the points were summed up for each respondent (range 0 to 7). Based on FAS distribution, the respondents were divided into two categories: low FAS (0–4 points; <25th quartile) and higher FAS (5–7 points). In the present study, the value of Cronbach’s alpha for the FAS (with four components) of 0.634 can be interpreted as sufficient internal consistency of the scale [4,42] as mentioned above (see Section 2.4) and it was a higher result than for the Polish adaptation of the scale reported by Mazur (0.497) [44].

### 2.6. Data Analysis

The normality of the distribution of continuous variables was assessed with the Shapiro–Wilk test. For continuous variables, the data were presented as means with 95% confidence intervals (95% CIs) or standard deviations (SDs), while categorical variables were presented as a sample percentage (%). As the NKS (in points) was not normally distributed, the differences between the means of two independent groups (educated vs. control) were verified with a Mann–Whitney test, the means of two dependent groups (baseline vs. follow-up) were compared using a Wilcoxon test. The percentage distribution of categorical variables was determined with a chi-squared test with Yates’ correction when necessary.

The following three groups of respondents were considered: (i) with no increase in the nutrition knowledge score, (ii) with an increase in the nutrition knowledge score, (iii) with an increase in the nutrition knowledge score ≥4 points, note: this group was also included in (ii). These three respondent groups (and cutoffs in points) were chosen a priori as optimal for comparison based on the (percentage) distribution of the changes in the NKS after three and nine months and with the assumption that the sample size was adequate for the statistical analysis.

To determine an association between nutrition knowledge and socioeconomic factors (gender, residence, FAS), logistic regression modelling was used to assess the chance of no increase in the nutrition knowledge score after a 3 month or 9month follow-up in respect to the baseline. As a reference, two categories were considered, separately: (i) any increase in the nutrition knowledge score, (ii) an increase in the nutrition knowledge score ≥4 points; the modelled category was no increase. The odds ratios (ORs) and 95% CI were then calculated. A crude model and a model with an adjustment for gender, age (years), residence (urban, rural) and FAS (three categories: 0–4 points, 5–6 points, 7 points) were created, excluding the modelled variable from the confounders set, respectively. The significance of ORs was assessed by Wald’s statistics. The Statistica software package (version 12.0 PL, StatSoft Inc., Tulsa, OK, USA, StatSoft, Krakow, Poland) was used for all data analyses. Statistical significance (*p*) was considered at three levels: <0.05, <0.01, <0.001.

## 3. Results

### 3.1. Changes in Nutrition Knowledge Score Three Months after Education

In the educated group, the increase in the NKS in the total sample and all subgroups (girls, boys, rural residence, urban residence, low FAS, higher FAS) was found, on average by 2.0 to 3.2 points (Table 2 and Figure 3). In the control group, the increase in the NKS averaged 0.6 to 0.7 points in all subgroups except for the low FAS subgroup (*p* = 0.08). In the total sample and all subgroups, the increase in the NKS was significantly higher in the educated group than in the control group, on average by 1.4 to 2.7 points (all *p* < 0.001).

### 3.2. Changes in Nutrition Knowledge Score Nine Months after Education

In the educated group, the increase in the NKS in the total sample and all subgroups (girls, boys, rural residence, urban residence, low FAS, higher FAS) was found to be, on average, by 1.6 to 3.2 points (Table 2 and Figure 3). In the control group, the increase in the NKS averaged 1.0 to 2.1 points in all subgroups. The increase in the NKS was significantly higher in the educated group than in the control group in rural residents (by 2.2 points, *p* < 0.001) and in the total sample (by 0.4, *p* < 0.05), while there was no significant difference in girls, boys, urban residents and teens with low or higher FAS.

### 3.3. The Chance of No Increase in Nutrition Knowledge Score Three Months after Education

In the educated group, the chance of no increase in the NKS was higher in urban residents than in rural residents, considering both any increase (adjusted OR 2.75, *p* < 0.01) and an increase by ≥4 points (adjusted OR 3.63, *p* < 0.001; Table 3; Table 4) as a reference. A similar association was found in the crude model. In the control group, the chance of no increase in the NKS was higher in the low FAS group (adjusted OR 2.67, *p* < 0.05) than in the higher FAS group considering any increase as a reference.

### 3.4. The Chance of No Increase in Nutrition Knowledge Score Nine Months after Education

In the educated group, the chance of no increase in the NKS was higher in urban residents (adjusted OR 2.99, *p* < 0.001) than in rural residents considering an increase by ≥4 points as a reference. A similar association was found in the crude model. In the control group, the chance of no increase in the NKS was lower in urban residents than in rural residents considering any increase (crude OR 0.46, *p* < 0.05) and an increase by ≥4 points (crude OR 0.30, *p* < 0.01) as a reference; however, the association disappeared after adjustment for confounders.

## 4. Discussion

Considering the total sample, multi-component education improved the nutrition knowledge of teenagers both three and nine months after education. This effect was visible after three months, regardless of sex, place of residence or family affluence, but weakened after nine months, with the exception of rural residents.

The study showed that the implemented multi-component education related to healthy eating, physical activity, culinary experiments and food safety in the long term has improved the nutrition knowledge of teenagers. Considering the total sample, the education program turned out to be effective after both three and nine months. A significant increase in the nutrition knowledge score was noted—in the educated group on average by 2.4 and 2.2 points after three and nine months, respectively. The nutrition knowledge score was also increased in the control group, but the average increase was significantly greater in the educated group than in the control group (by 1.8 and 0.4 points for difference after 3 and 9 months, respectively). The increase in nutrition knowledge in the control group may be attributed to regular school education and out-of-school influences, while in the educated group it was undoubtedly the result of the implemented education program together with school and out-of-school influences. Some previous studies have also evaluated the effectiveness of multi-component intervention models of nutritional and lifestyle education, but the follow-up period from these interventions was shorter and ranged from three to six months [45,46,47]. Only Zhou et al. [17] studied the effectiveness of pre-/post-intervention within a nine-month follow-up. In contrast to the current study, knowledge scores in the intervention group were significantly lower after the three- and nine-month follow-ups than immediately after the intervention [17].

The studies mentioned above included multi-component intervention packages and were addressed to children and adolescents (Iranian 12–16 years old; Indian 8–18 years old; Chinese 11–14 years old) and their parents and teachers [17,45,46,47], so they were differently designed in comparison with the current study. In the present study, the education program was related to 11–12-year-old teenagers and was not supported by parents or teachers during education and follow-up. Due to the positive result of the current education program in a nine-month follow-up and the failure of the program by Zhou et al. [17], the current results confirm that the proposed model of education addressed only to 11–12-year-old teens was effective and can be widely recommended. 

When considering the residence, the results clearly showed that the increase in nutrition knowledge score was much higher among students living in rural areas than those living in urban areas. For rural teens, these differences remained statistically significant also after nine months from the intervention, being similar to those after three months (average difference was 2.2 and 2.5 points, respectively). Moreover, the chance of the lack of an educational effect, taking into account confounders and regardless of the referent category, was approximately three times greater in urban teens than in rural teens. The positive effect of educational intervention on nutrition knowledge or practice has been shown by several studies of rural residents [18,48,49,50]. However, the mentioned studies covered only the rural teen populations, thus the effectiveness of nutrition education refers only to them. The current study was more broadly designed and compared the same model of education in different places of residence to assess whether it is a factor influencing the outcome of nutrition knowledge. The findings from the current study are in line with a cross-sectional study carried out in the federal state of Tyrol in western Austria [51], which found associations of higher nutrition knowledge in Austrian students from rural schools and better use of weekly nutrition education classes in comparison to students from urban schools.

It may be speculated that the urban environment provides young people with more stimuli and creates various opportunities to modulate their knowledge, both increasing or decreasing it. Frequent contact with many people, both direct or via social media, and the appearance of much news in the public spaces may weaken previously gained knowledge that has not been consolidated or change it based on false evidence. So-called “fake” news related to diet and health is especially frequently distributed via social media and the Internet through idols and influencers [52,53,54]. Such news may be uncritically and easily adopted by a young audience who do not have the competence to verify it with scientific evidence. There is another possible explanation for the differences between rural and urban teens, although this is only an observation unsupported by scientific investigation. The authors noted a greater interest in the educational program and enthusiasm for the activities that took place in academic centres among rural teens. Moreover, students from rural schools were strongly supported by their teachers to increase student interest, while teachers from urban schools were less (or not) interested in supporting their students. Therefore, it is possible to conclude from the current research, supported by previous results, that rural teens are more sensitive to multi-component nutrition education, while special attention should be paid to urban teens. To achieve a long-lasting educational effect in urban teens, additional activities supporting the previously gained knowledge should be considered. In the authors’ opinions, nutrition education programs should be designed to provide additional simple, supportive education a few months after basic education has been completed.

As far as sex subgroups are concerned, the educational effect was similar in girls and boys. The average difference in the increase in nutrition knowledge score, three months after education, was higher in girls/boys who attended the education program than in those out of the program; however, nine months after education, this significance disappeared both in girls and boys. Moreover, the chance of the lack of an educational effect, taking into account confounders and regardless of the referent category, was similar in girls and boys. This finding is in line with a study by Lai Yeung [55]. The author found that 11–16-year-old boys and girls from Hong Kong showed a similar level of nutrition knowledge and, moreover, a similar consumption of vegetables, fruit, milk and bread. In contrast to the current findings and Lai Yeung [55], most studies have found that girls have healthier eating habits and higher nutrition knowledge than boys [56,57,58,59]. Cross-cultural studies indicate that women focus on healthy eating choices, e.g., avoiding fat-rich foods and salt as well as consuming more fruits and fibre, for better weight control [57,60]. Lipowska et al. [58] suggested that even five-year-old girls are probably aware of the social importance of diet and, thus, internalize healthier eating habits. Altogether, this indicates that while gender plays a significant role in food choices and preferences, the education related to knowledge of nutrition and lifestyle is similarly effective in girls and boys.

Regarding family affluence, the educational effect was similar in teenagers with low and higher family affluence. As was described above with respect to sex, among teenagers with low and higher family affluence, a similar improvement in nutrition knowledge score three months after education was found to have no significant improvement after nine months. Furthermore, the chance of the lack of an educational effect was similar in both teenager groups (with low and higher family affluence). This finding suggests that family affluence is not a factor in shaping long-term maintenance of the nutrition knowledge among Polish teenagers and this factor can be excluded from the confounders set when designing educational programs. It is difficult to compare the current findings with others, because the present study was designed as an intervention with a control group in which the change in nutrition knowledge was a study outcome. Some studies involving adolescents from Europe, Canada and Israel included dietary habits as a study outcome and showed that a high level of inequality in daily food consumption was associated with family wealth [61,62,63]. However, due to the cross-sectional design of those studies, the changes in nutrition knowledge related to family affluence cannot be interpreted. Further studies of teenagers from other European countries should be carried out to confirm (or not) that family affluence has no impact on the improvement and long-term maintenance of teenagers’ nutrition knowledge.

### Strengths and Limitations

The main advantage of this study was the relatively large sample size (464 teenagers) subjected to a 9 month follow-up. A very rigorous selection of the sample was applied, each participant had to take part in lectures and workshops within all five topics of the education program lasting three weeks; students who missed even one lecture or workshop were excluded. Data collection and all measurements were taken by well-trained researchers with the same type of equipment in all scientific centres to minimize inter-centre differences. Data related to nutrition knowledge and socioeconomic variables were collected with a food frequency questionnaire (SF-FFQ4PolishChildren), which has a known quality and can be recommended to evaluate dietary and lifestyle behaviours among children and adolescents [37].

The main limitation is a lack of random allocation of subjects to the educated and control groups. Unfortunately, the authors were unable to apply the random approach for several reasons. First, for organizational reasons, the authors wanted to choose schools located at a convenient distance from the academic centres (up to 50 km). Second, many school principals (surprisingly) did not permit their school to participate in the study. Third, many of the schools had previously participated in other nutrition- or health-related education programs, so they could not be included in our study. There was an element of randomness in the current study since assigning classes to the educated or control group was accidental. Since the study sample is not representative on a national level, study findings do not provide the full picture of the educational effect in 11–12-year-old teens across Poland. The sample could also be biased due to the lack of permission given by the school, teacher or parent to enrol a child in an educational program. However, there is currently no basis to speculate on whether the educational effect would be higher or lower without this bias.

## 5. Conclusions

The results suggest that, in general, the “ABC of Healthy Eating” program is a good model for improving teenagers’ nutrition knowledge. Taking into consideration the socioeconomic context, the “ABC of Healthy Eating” program improved the nutrition knowledge of teenagers in the short term regardless of sex, residence or family affluence, but in the long term, this effect was visible only in rural residents. This suggests that although the findings are not representative on the national level, rural teens are more sensitive to multi-component nutrition education, while special attention should be paid to urban teens if nutrition education is being developed and implemented.

Future studies should focus on a comprehensive understanding of barriers attenuating the long-term effectiveness of multi-component, nutrition education in urban teenage residents and, next, developing and testing a special path of nutrition education addressed to urban teens. A simple temporary solution is proposed to design nutrition education programs containing additional simple, supportive education a few months after basic education has been completed. Such a design should strengthen the effects of nutrition education and boost its long-term maintenance in urban teens and bring additional benefits to other socioeconomic groups of teenagers.

## Figures and Tables

**Figure 1 nutrients-13-01661-f001:**
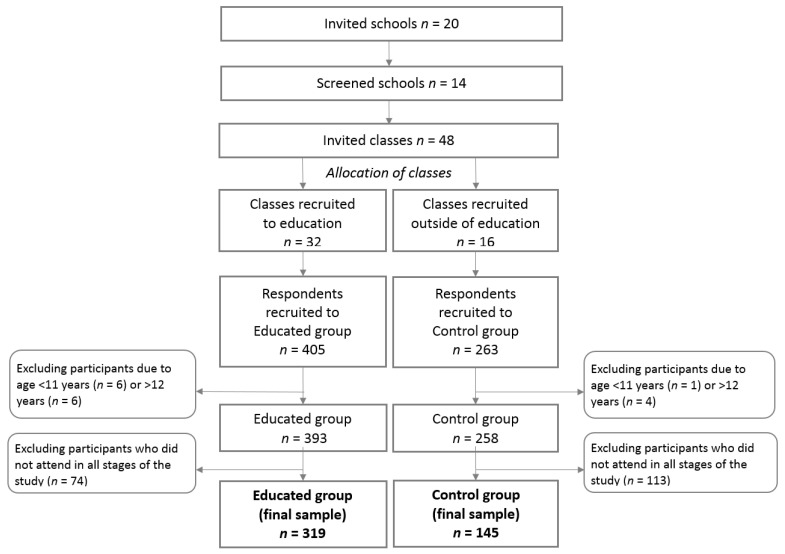
Flow chart of sample collection and study design.

**Figure 2 nutrients-13-01661-f002:**
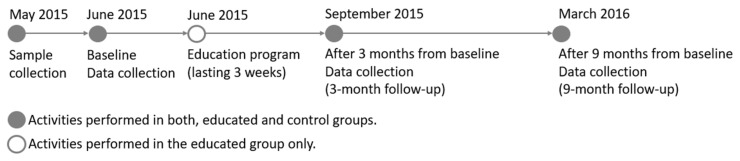
Timeline and activities of the ABC-HEat program.

**Figure 3 nutrients-13-01661-f003:**
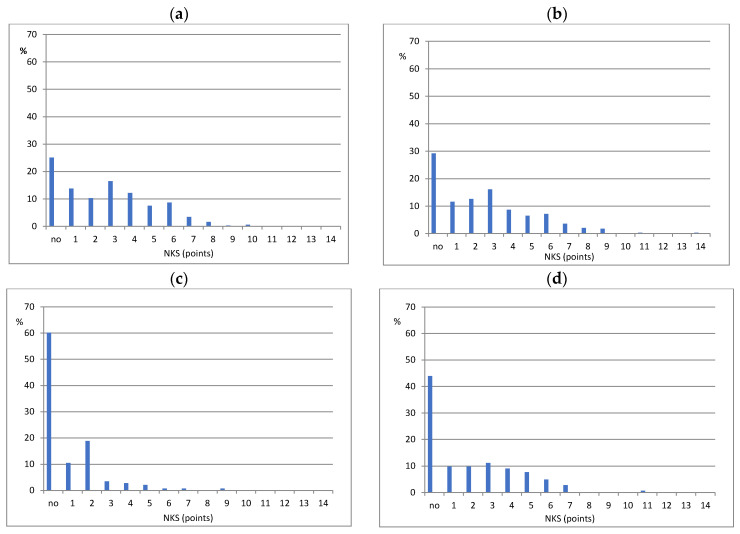
Sample distribution (%) by changes in nutrition knowledge score (NKS) after three months and nine months. Notes: no—means no increase in NKS; (**a**) educated group after 3 months; (**b**) educated group after 9 months; (**c**) control group after 3 months; (**d**) control group after 9 months.

**Table 1 nutrients-13-01661-t001:** Sample baseline characteristics and changes in nutrition knowledge score (NKS) three months and nine months after education (% or mean and standard deviation, SD).

Variables	Total	Educated	Control	*p*-Value
Sample size	464	319	145	
Sample percentage	100.0	100.0	100.0	
Age (years), mean (SD)	11.9 (0.3)	11.9 (0.3)	12.0 (0.2)	0.20
Gender, *n* (%)				0.16
Girls	248 (53.4)	178 (55.8)	70 (48.3)	
Boys	216 (46.6)	141 (44.2)	75 (51.7)	
Residence, *n* (%)				0.06
Urban	302 (65.1)	217 (68.0)	85 (58.6)	
Rural	162 (34.9)	102 (32.0)	60 (41.4)	
Family Affluence Scale, *n* (%)				
Low	117 (25.3)	79 (24.9)	38 (26.2)	0.86
Higher	345 (74.7)	238 (75.1)	107 (73.8)	
NKS (points), mean (SD)				
Baseline	6.0 (2.7)	6.1 (2.6)	5.5 (2.7)	<0.05
After 3 months	7.8 (3.0)	8.5 (2.8)	6.2 (2.9)	<0.001
After 9 months	7.9 (3.0)	8.3 (3.1)	7.2 (2.7)	<0.001
Change after 3 months	1.8 (2.7)	2.4 (2.8)	0.6 (1.9)	<0.001
Change after 9 months	2.0 (3.2)	2.1 (3.4)	1.7 (2.6)	<0.05
No increase in NKS after 3 months, *n* (%)	166 (35.9)	80 (25.1)	86 (60.1)	<0.001
Increase in NKS by 1–3 points after 3 months, *n* (%)	177 (38.3)	130 (40.7)	47 (32.9)	
Increase in NKS by ≥4 points after 3 months, *n* (%)	119 (25.8)	109 (34.2)	10 (7.0)	
No increase in KNS after 9 months, *n* (%)	156 (33.8)	93 (29.2)	63 (44.0)	<0.01
Increase in KNS by 1–3 points after 9 months, *n* (%)	172 (37.2)	128 (40.1)	44 (30.8)	
Incerase in KNS by ≥4 points after 9 months, *n* (%)	134 (29.0)	98 (30.7)	36 (25.2)	

Sample size may vary in variables due to missing data. Family Affluence Scale (low: 0–4 points, higher: 5–7 points). Nutrition knowledge score range: 0–18 points. *p*–value significance level of Mann–Whitney test (for means) or hi^2^ test, with Yates’ correction when necessary (for percentage distribution).

**Table 2 nutrients-13-01661-t002:** Changes in nutritional knowledge score (points) three months and nine months after education by socioeconomic factors (mean and 95% confidence interval, 95% CI).

Variables	Baseline	After 3 Months	After 9 Months	Change after 3 Months	*p*-Value	Change after 9 Months	*p*-Value
Total sample							
educated	6.1 (5.9; 6.4)	8.5 (8.2; 8.8)	8.3 (7.9; 8.6)	2.4 (2.0; 2.7)	<0.001	2.1 (1.7; 2.5)	<0.001
control	5.5 (5.1; 6.0)	6.2 (5.7; 6.6)	7.2 (6.8; 7.6)	0.6 (0.3; 0.9)	<0.001	1.7 (1.2; 2.1)	<0.001
Difference	0.6	2.3	1.1	1.8		0.4	
*p*-value	<0.05	<0.001	<0.001	<0.001		<0.05	
Girls							
educated	6.6 (6.2; 7.0)	8.9 (8.5; 9.4)	8.6 (8.2; 9.1)	2.4 (1.9; 2.8)	<0.001	2.0 (1.6; 2.5)	<0.001
control	5.7 (5.0; 6.4)	6.3 (5.6; 7.0)	7.3 (6.7; 8.0)	0.6 (0.2; 1.0)	<0.01	1.7 (1.0; 2.3)	<0.001
Difference	0.9	2.6	1.3	1.8		0.3	
*p*-value	<0.05	<0.001	<0.01	<0.001		0.15	
Boys							
educated	5.6 (5.1; 6.0)	7.9 (7.5; 8.4)	7.8 (7.3; 8.3)	2.3 (1.9; 2.8)	<0.001	2.2 (1.6; 2.8)	<0.001
control	5.4 (4.8; 6.0)	6.1 (5.4; 6.7)	7.1 (6.5; 7.7)	0.6 (0.1; 1.1)	<0.05	1.6 (1.0; 2.3)	<0.001
Difference	0.2	1.8	0.7	1.7		0.6	
*p*-value	0.61	<0.001	<0.05	<0.001		0.10	
Urban							
educated	6.3 (5.9; 6.6)	8.2 (7.8; 8.6)	7.9 (7.4; 8.3)	2.0 (1.6; 2.3)	<0.001	1.6 (1.2; 2.0)	<0.001
control	4.7 (4.2; 5.2)	5.3 (4.7; 5.8)	6.8 (6.3; 7.4)	0.6 (0.2; 0.9)	<0.01	2.1 (1.5; 2.7)	<0.001
Difference	1.6	2.9	1.1	1.4		−0.5	
*p*-value	<0.001	<0.001	<0.01	<0.001		0.27	
Rural							
educated	5.9 (5.4; 6.4)	9.1 (8.6; 9.6)	9.1 (8.5; 9.6)	3.2 (2.6; 3.7)	<0.001	3.2 (2.5; 3.9)	<0.001
control	6.8 (6.0; 7.5)	7.4 (6.7; 8.2)	7.7 (7.0; 8.5)	0.7 (0.2; 1.3)	<0.05	1.0 (0.3; 1.6)	<0.01
Difference	−0.9	1.7	1.4	2.5		2.2	
*p*-value	0.06	<0.001	<0.01	<0.001		<0.001	
Low FAS							
educated	5.7 (5.1; 6.2)	8.7 (8.1; 9.3)	8.4 (7.8; 9.0)	3.0 (2.4; 3.6)	<0.001	2.7 (2.0; 3.5)	<0.001
control	4.4 (3.6; 5.2)	4.8 (4.0; 5.5)	6.4 (5.6; 7.2)	0.3 (0.0; 0.7)	0.08	2.0 (1.0; 3.0)	<0.001
Difference	1.3	3.9	2.0	2.7		0.7	
*p*-value	<0.05	<0.001	<0.001	<0.001		0.11	
Higher FAS							
educated	6.3 (6.0; 6.7)	8.5 (8.1; 8.8)	8.2 (7.8; 8.6)	2.1 (1.8; 2.5)	<0.001	1.9 (1.5; 2.3)	<0.001
control	5.9 (5.4; 6.5)	6.7 (6.1; 7.2)	7.5 (7.0; 8.0)	0.7 (0.3; 1.1)	<0.001	1.5 (1.0; 2.0)	<0.001
Difference	0.4	1.8	0.7	1.4		0.4	
*p*-value	0.16	<0.001	<0.05	<0.001		0.10	

Nutrition knowledge score (range: 0–18). FAS indicates Family Affluence Scale (low: 0–4 points, higher: 5–7 points). Difference calculated as the absolute difference between the educated vs. control group. Change calculated as the difference between follow-up vs. baseline within one group (educated or control). *p*-value—significance level for difference (Mann–Whitney test) or change (Wilcoxon test).

**Table 3 nutrients-13-01661-t003:** Sample distribution (%) by the change in nutrition knowledge score and socioeconomic factors three months and nine months after education.

Variables	After 3 Months												After 9 Months											
	No Increase				Any Increase				Increase ≥4 Points				No Increase				Any Increase				Increase ≥4 Points			
	Total	Edu	Con	*p*-Value	Total	Edu	Con	*p*-Value	Total	Edu	Con	*p*-Value	Total	Edu	Con	*p*-Value	Total	Edu	Con	*p*-Value	Total	Edu	Con	*p*-Value
Sample size	166	80	86		296	239	57		119	109	10		156	93	63		306	226	80		134	98	36	
Sample percentage	100.0	48.2	51.8		100.0	80.7	19.3		100.0	91.6	8.4		100.0	59.6	40.4		100.0	73.9	26.1		100.0	73.1	26.9	
Gender				0.42				0.16				0.17				0.36				0.41				0.25
Girls	53.6	57.5	50.0		53.0	55.2	43.9		55.5	57.8	30.0		51.3	54.8	46.0		54.6	56.2	50.0		48.5	52.0	38.9	
Boys	46.4	42.5	50.0		47.0	44.8	56.1		44.5	42.2	70.0		48.7	45.2	54.0		45.4	43.8	50.0		51.5	48.0	61.1	
Residence				<0.01				0.56				0.49				<0.001				0.95				<0.05
Urban	70.5	82.5	59.3		62.2	63.2	57.9		55.5	56.9	40.0		64.1	75.3	47.6		65.4	65.0	66.3		56.7	50.0	75.0	
Rural	29.5	17.5	40.7		37.8	36.8	42.1		44.5	43.1	60.0		35.9	24.7	52.4		34.6	35.0	33.8		43.3	50.0	25.0	
FAS																								
Low	24.1	17.5	30.2		22.0	23.4	16.1		24.4	25.7	10.0		18.1	15.2	22.2		24.1	22.3	29.1		27.1	23.5	37.1	
Higher	75.9	82.5	69.8	0.08	78.0	76.6	83.9	0.31	75.6	74.3	90.0	0.47	81.9	84.8	77.8	0.37	75.9	77.7	70.9	0.29	72.9	76.5	62.9	0.18

Nutrition knowledge score (range: 0–18). FAS indicates Family Affluence Scale (low: 0–4 points, higher: 5–7 points). Edu indicates—educated group, Con—control group. *p*-value—significance level of chi-squared test with Yates’ correction.

**Table 4 nutrients-13-01661-t004:** Odds ratios (95% Confidence Intervals) for no increase in nutrition knowledge score three months and nine months after education by socioeconomic factors.

Variables	Models	After 3 Months		After 9 Months	
		No Increase (Ref.: Any Increase)	No Increase (Ref.: Increase by ≥4 Points)	No Increase (Ref.: Any Increase)	No Increase (Ref.: Increase by ≥4 Points)
Girls (ref.: boys)					
educated	Crude	1.10 (0.66, 1.83)	0.99 (0.55, 1.78)	0.95 (0.58, 1.54)	1.12 (0.63, 1.99)
	Adjusted	1.05 (0.62, 1.77)	0.85 (0.46, 1.59)	0.92 (0.56, 1.51)	1.02 (0.56, 1.87)
control	Crude	1.28 (0.65, 2.52)	2.33 (0.56, 9.80)	0.85 (0.35, 2.05)	1.34 (0.58, 3.12)
	Adjusted	1.29 (0.64, 2.58)	2.36 (0.52, 10.59)	0.85 (0.43, 1.69)	1.31 (0.54, 3.19)
Urban (ref.: rural)					
educated	Crude	2.75 ** (1.45, 5.19)	3.57 *** (1.78, 7.16)	1.64 (0.95, 2.83)	3.04 *** (1.64, 5.65)
	Adjusted	2.75 ** (1.45, 5.20)	3.63 *** (1.80, 7.31)	1.66 (0.96, 2.89)	2.99 *** (1.60, 5.59)
control	Crude	1.06 (0.53, 2.10)	2.19 (0.56, 8.46)	0.46 * (0.23, 0.92)	0.30 ** (0.12, 0.76)
	Adjusted	0.77 (0.35, 1.66)	1.03 (0.19, 5.57)	0.48 (0.23, 1.01)	0.38 (0.14, 1.05)
Low FAS (ref.: higher)					
educated	Crude	0.69 (0.36, 1.33)	0.61 (0.30, 1.27)	0.62 (0.33, 1.20)	0.59 (0.28, 1.23)
	Adjusted	0.74 (0.38, 1.43)	0.67 (0.32, 1.42)	0.66 (0.34, 1.27)	0.67 (0.31, 1.46)
control	Crude	2.26 (0.96, 5.33)	3.90 (0.46, 33.30)	0.70 (0.32, 1.51)	0.48 (0.19, 1.21)
	Adjusted	2.67 * (1.08, 6.63)	3.73 (0.39, 35.25)	0.83 (0.37, 1.86)	0.68 (0.25, 1.86)

Sample size may vary in variables due to missing data. FAS indicates Family Affluence Scale (low: 0–4 points, higher: 5–7 points). Adjusted model: odds ratios adjusted for confounders (at follow-up): gender (girls, boys), age (years), residence (urban, rural), FAS (three categories: 0–4 points, 5–6 points, 7 points), excluding the modelled variable from the confounders’ set, respectively. Statistically significant (Wald’s statistics): * *p*-value < 0.05, ** *p* < 0.01, *** *p* < 0.001.

## Data Availability

Due to ethical restrictions and participant confidentiality, data cannot be made publicly available. However, data from the ABC of Healthy Eating study are available upon request, for researchers who meet the criteria for access to confidential data. Data requests can be sent to the ABC of Healthy Eating study coordinator (Jadwiga Hamulka).

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
