# Peer review of "Sustainability of a Multi-Component Education Program (ABC of Healthy Eating) after Three Months and Nine Months: The Socioeconomic Context in Improving Nutrition Knowledge in Polish Teenagers"

_nutrients, 2021, doi:10.3390/nu13051661_

Round 1

Reviewer 1 Report

Dear Authors, 

This is a very well written paper describing the results of a nutrition education program for teens. Minor comments for your consideration.

  1. Could the authors provide additional information in the methods section in regards to subject selection -  although it is noted that previous papers have published this information.
  2. Is the study size powered correctly to determine change? Could this information be provided.
  3. Could the authors explore and/or propose additional possibilities for the difference in rural and urban after 9 months follow up.
  4. Could the authors address additional limitations including sample bias given the school, teacher and parent permissions. Are the findings representative of all teens across Poland?

Author Response

                                                   Malgorzata Kostecka

Faculty of Food Science and Biotechnology

University of Life Sciences

15 Akademicka Street, 20-960 Lublin, Poland

Authors’ Response to the Reviewers’ Comments

Journal:          Nutrients

Manuscript:   nutrients-1195818

Title:                Sustainability of a multi-component education program (ABC of Healthy Eating) after three months and nine months: the socioeconomic context in improving nutrition knowledge in Polish teenagers

Authors:          Lidia Wadolowska, Malgorzata Kostecka* , Joanna Kowalkowska, Marta Jeruszka-Bielak, Marzena Tomaszewska, Anna Danielewicz, Jadwiga Hamulka

Article type:    research article

7th May 2021

Dear Roy Zhang, Dear Reviewers,

Thank you for revising our manuscript entitled ‘Sustainability of a multi-component education program (ABC of Healthy Eating) after three months and nine months: the socioeconomic context in improving nutrition knowledge in Polish teenagers’.

We greatly appreciate the time and efforts to review our manuscript and we agree that the proposed changes will contribute to the improvement of our manuscript. We have addressed all issues indicated in reviews, and we believe that the revised version can meet the journal publication requirements.

Please find our responses to the Reviewers’ comments attached. The changes made in the text are highlighted in blue.

Yours Sincerely,

Malgorzata Kostecka

Response to Comments from Reviewer #1

Reviewer’s Comment

Response/amendment

Location (lines)

This is a very well written paper describing the results of a nutrition education program for teens. Minor comments for your consideration.

Thank you for an insightful review and all suggestions. We agree the suggested changes will contribute to the improvement of our paper. We hope you will find our improvements appropriate and comprehensive.

1.      Could the authors provide additional information in the methods section in regards to subject selection -  although it is noted that previous papers have published this information.

The methods section has been reformatted to better present information in regards to subject selection, and also some details and Figure S1 (in supplementary material) were added.

In respect to educational program – related data have been presented in Tables S1 and S2: (S1) Topics and details of the education program; (S2) Questions and correct answers (scored with 1 point) regarding an assessment of nutrition knowledge.  Unfortunately, the file with supplementary materials was not visible on  the platform, even though it was loaded– we apologize for this inconvenience.  

103-128, Figure S1

2.      Is the study size powered correctly to determine change? Could this information be provided.

Thank you this important question. The sample size included in the study (educated/control group 319/145) exceeded the minimum sample size calculated (200/104) – based on our previous paper [Hamulka et al. 2018; DOI 10.3390/nu10101439], related information was added in the text.

133-142

3.      Could the authors explore and/or propose additional possibilities for the difference in rural and urban after 9 months follow up.

Thank you for your insightful feedback. The next explanation for the difference between rural and urban teens in nutrition knowledge was added in the text (discussion section).

316-320

4.      Could the authors address additional limitations including sample bias given the school, teacher and parent permissions. Are the findings representative of all teens across Poland?

Thank you for an insightful suggestion.  Additional limitations including sample bias were discussed in the strengths and limitations sub-section.

368-371

Reviewer 2 Report

I need more information on the NKS. Provide distributional characteristics, like means and sd's. Am I correct to assume that the mean score is something like 6? Is this not extremely low, with 18 items? How is this possible? Provide the results from factor and reliability analyses.

Where does the NKS categorization of => 4 points come from?

Why not use t-tests, anova or manova to test the developments (in means, and not categories)?

Provide the results from factor and reliability analyses regarding the FAS.

The right part of tables 2 and 3 are missing.

Are there any differences between the original sample and the final sample?

Author Response

Journal:          Nutrients

Manuscript:   nutrients-1195818

Title:                Sustainability of a multi-component education program (ABC of Healthy Eating) after three months and nine months: the socioeconomic context in improving nutrition knowledge in Polish teenagers

Authors:          Lidia Wadolowska, Malgorzata Kostecka* , Joanna Kowalkowska, Marta Jeruszka-Bielak, Marzena Tomaszewska, Anna Danielewicz, Jadwiga Hamulka

Article type:    research article

7th May 2021

Dear Roy Zhang, Dear Reviewers,

Thank you for revising our manuscript entitled ‘Sustainability of a multi-component education program (ABC of Healthy Eating) after three months and nine months: the socioeconomic context in improving nutrition knowledge in Polish teenagers’.

We greatly appreciate the time and efforts to review our manuscript and we agree that the proposed changes will contribute to the improvement of our manuscript. We have addressed all issues indicated in reviews, and we believe that the revised version can meet the journal publication requirements.

Please find our responses to the Reviewers’ comments attached. The changes made in the text are highlighted in blue.

Yours Sincerely,

Malgorzata Kostecka

Response to Comments from Reviewer #2

Reviewer’s Comment

Response/amendment

Location (lines)

Thank you for an insightful review and all suggestions. We agree the suggested changes will contribute to the improvement of our paper. We hope you will find our improvements appropriate and comprehensive.

1.      I need more information on the NKS. Provide distributional characteristics, like means and sd's. Am I correct to assume that the mean score is something like 6? Is this not extremely low, with 18 items? How is this possible? Provide the results from factor and reliability analyses.

In regards to the NKS expressed in points, means and SDs were added in Table 1 to better describe this variable, however, means with 95%CI for education and control group have been presented in Table 2 (except data for total sample).

You are right, the mean of the NKS for the total sample at baseline was 6.0 points and gained on average by 1.8 points after 3 months and 2.0 points after 9 months. In our opinion, the average of 6 points in the range of 0-18 points is not surprising in relation to nutritional knowledge, which is strongly influenced e.g. by many myths coming from, among others, social media and TV (it was discussed in the text), moreover, correct nutrition knowledge with high points above 75 percentile is rare. Furthermore, slightly higher mean for the NKS (7.2 points) in 11-15-years-old adolescents (n=626) was found when the internal compatibility of a short-form, multicomponent dietary questionnaire (SF-FFQ4PolishChildren) was assessed [Kowalkowska et al. Nutrients 2019; doi:10.3390/nu11122929].

Results of the reliability analysis (with the Cronbach's alpha) for the NKS were added in the text.

Results of factor analysis for the NKS were not included in the text. We have made an attempt to perform a factor analysis for the components of NKS, but results were not satisfactory, for factors obtained in the analysis (with eigenvalues above 1.0), with the NKS items as input variables, the total explained variance was 50.8%.

According to your comment, we improved the text by adding more explanations (section 2.4). We hope you will find it relevant.

Tab.1, 170-172, 175-181

2.      Where does the NKS categorization of => 4 points come from?

The categorization of the NKS has been carried out a priori – three respondent groups and cut-offs in points were chosen as optimal for comparison on the basis of the (percentage) distribution of the changes in the NKS after three and nine months, and also considering sample size being adequate to the statistical analysis. Such explanation was added in the text.

212-215

3.      Why not use t-tests, anova or manova to test the developments (in means, and not categories)?

The NKS expressed in points was not normally distributed (tested with the Shapiro-Wilk test), therefore non-parametric tests were used, i.e. Mann-Whitney test for independent samples (to compare educated and control group) and Wilcoxon test for dependent samples (to  assess the follow-up changes). The categorized NKS was compared with a chi2 test with Yates' correction when necessary (results presented in Table 3) and a logistic regression with Wald’s statistics (results presented in Table 4). The text was clarified respectively.

203-205

4.      Provide the results from factor and reliability analyses regarding the FAS.

Thank you for an insightful suggestion.  Results of the reliability analysis (with the Cronbach's alpha) for the FAS were added in the text.

Results of factor analysis for the FAS were not included in the text. In the present study, we found a unifactorial structure of the scale (FAS), the total explained variance was 48.4%, eigenvalue was 1.93. Furthermore, we used the scale adopted by the Polish team of the HBSC study [Mazur, J. Family Affluence Scale—validation study and suggested modification. Hygeia Public Health 2013, 48, 211–217]. Similarly to our result, Mazur (2013) reported a unifactorial structure of the scale (FAS).

The new Table S3 has been added in the supplementary material with scoring response details for the calculation of the FAS.

193-196

Table S3

5.      The right part of tables 2 and 3 are missing.

We are very sorry for your inconvenience – both Tables 2 and 3 were correctly and fully displayed in DOXC format, but  Table 2 was partly missed in PDF format. We have improved it.

6.      Are there any differences between the original sample and the final sample?

Thank you this important question. There were no differences, except age, between the initial sample (n=651) and the final sample (n=464); note: the initial sample consisted of participants who attended all stages of the study (464) and participants who not attended all stages of the study (187), after excluding participants with inadequate age (<11 or >12 years).

Table A (attached to the response) provides the result of the comparison between the initial sample and the final sample.

Table A. Characteristics of participants in the initial# and the final sample (% or mean and standard deviation SD).

Variables

Total

Educated

Control

Initial sample

Final sample

p-value

Initial sample

Final sample

p-value

Initial sample

Final sample

p-value

Sample size

651

464

393

319

258

145

Sample percentage

100.0

100.0

60.4

68.8

39.6

31.2

Age (years), mean (SD)

11.9 (0.3)

11.9 (0.3)

<0.05

11.9 (0.3)

11.9 (0.3)

0.41

11.9 (0.3)

12.0 (0.2)

<0.05

Gender, n (%)

Girls

53.8

53.4

0.97

56.0

55.8

0.98

50.4

48.3

0.76

Boys

46.2

46.6

44.0

44.2

49.6

51.7

Residence, n (%)

Urban

68.4

65.1

0.28

68.4

68.0

0.97

68.2

58.6

0.07

Rural

31.6

34.9

31.6

32.0

31.8

41.4

Family Affluence Scale, n (%)

Low

24.0

25.3

0.67

21.7

24.9

0.36

27.5

26.2

0.87

Higher

76.0

74.7

78.3

75.1

72.5

73.8

NKS (points), mean (SD)

Baseline

5.9 (2.7)

6.0 (2.7)

0.94

6.2 (2.7)

6.1 (2.6)

0.92

5.6 (2.7)

5.5 (2.7)

0.76

After 3 months

7.7 (3.0)

7.8 (3.0)

0.60

8.4 (2.9)

8.5 (2.8)

0.60

6.3 (2.9)

6.2 (2.9)

0.67

After 9 months

7.8 (3.0)

7.9 (3.0)

0.48

8.2 (3.1)

8.3 (3.1)

0.89

7.1 (2.7)

7.2 (2.7)

0.65

Change after 3 months

1.7 (2.8)

1.8 (2.7)

0.52

2.2 (3.0)

2.4 (2.8)

0.66

0.7 (1.9)

0.6 (1.9)

0.90

Change after 9 months

1.9 (3.2)

2.0 (3.2)

0.77

2.1 (3.4)

2.1 (3.4)

0.92

1.6 (2.7)

1.7 (2.6)

0.96

Change in NKS after 3 months, n (%)

No increase

38.6

35.9

0.66

26.9

25.1

0.85

61.5

60.1

0.74

Increase by 1-3 points

36.0

38.3

39.2

40.7

29.7

32.9

Increase by ≥ 4 points

25.4

25.8

33.9

34.2

8.8

7.0

Change in NKS after 9 months, n (%)

No increase

34.4

33.8

0.98

29.9

29.2

0.97

42.2

44.0

0.94

Increase by 1-3 points

36.7

37.2

39.2

40.1

32.3

30.8

Increase by ≥ 4 points

28.9

29.0

30.9

30.7

25.5

25.2

In the initial sample, sample size may vary in variables due to missing data. #after exclusion participants with inadequate age (<11 years or >12 years). Family Affluence Scale (low: 0–4 points, higher: 5–7 points). Nutrition knowledge score range: 0-18 points. p–value significance level of Mann-Whitney test (for means) or chi2 test, with Yates' correction when necessary (for percentage distribution).
